# Spiral Evolution of Visual World Model: Reclaiming Autoregression from the Diffusion Era

## Abstract

Recent advances in video generation have been dominated by diffusion-based models, which produce high-quality, prompt-faithful sequences through holistic denoising. While this paradigm has achieved striking visual fidelity, it falls short for real-time, interactive applications that require frame-level responsiveness and causal coherence—cornerstones of practical world modeling. In this position paper, we advocate for a strategic return to autoregressive generation as the foundational architecture for building interactive world simulators. We argue that beyond offering faster inference, autoregressive models bring critical structural advantages: they naturally support predictive compression, enable causal disentanglement, and offer a more responsive mechanism for integrating control signals in dynamic settings. Unlike language-conditioned diffusion models, autoregression flexibly accommodates frame-wise control inputs such as camera motion and joint actions, making it ideally suited for agent-centric simulation. We further highlight emerging techniques and promising directions—including selective denoising, adaptive resolution, and postdictive coding—that address historical limitations of autoregression and unlock new levels of interactivity. We contend that embracing autoregression will be essential for developing practical, controllable, and truly intelligent world models.

## 1 Introduction

In recent years, diffusion-based models [1, 2] have taken center stage in video generation, praised for their ability to produce high-fidelity, prompt-aligned sequences through holistic denoising. This paradigm has fueled breakthroughs in creative and cinematic applications, where visual quality and global coherence are key. However, the very design that enables such visual excellence—global optimization across the entire video—also makes diffusion models inherently slow and computationally intensive. These limitations hinder their applicability in real-time, interactive settings.

Before the diffusion era, autoregressive models—often built on ConvRNN architectures [3, 4]—were the standard, as illustrated in Fig. 1. These models generated videos frame by frame, embracing a causal structure that naturally supported temporal reasoning and sequential feedback. Despite these strengths, they fell out of favor due to limited visual quality and rigid control mechanisms. Ironically, as the field shifts from offline synthesis to embodied intelligence and interactive simulation—domains where responsiveness, causal coherence, and multimodal control are paramount—the foundational strengths of autoregression are becoming increasingly relevant. The task ahead is not to revert to obsolete architectures but to re-envision autoregressive generation with modern advances, positioning it as the backbone of practical world models.

In this position paper, we argue that **video generation—particularly for world modeling—stands to benefit significantly from a renewed focus on autoregressive methods**. While diffusion-based

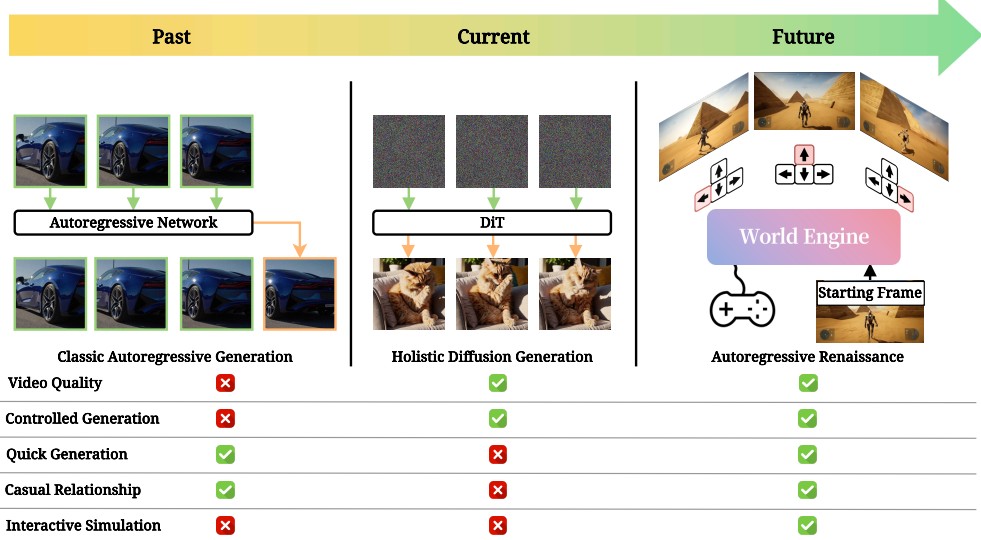

| | Classic Autoregressive Generation | Holistic Diffusion Generation | Autoregressive Renaissance |
|---|:---:|:---:|:---:|
| Video Quality | ❌ | ✅ | ✅ |
| Controlled Generation | ❌ | ✅ | ✅ |
| Quick Generation | ✅ | ❌ | ✅ |
| Casual Relationship | ✅ | ❌ | ✅ |
| Interactive Simulation | ❌ | ❌ | ✅ |

Figure 1: Illustration of the paradigm shift in video generation: we advocate revisiting the autoregressive paradigm as a foundation for building more powerful and interactive world models.

models excel at holistic scene synthesis and compositional generalization, they fall short in tasks demanding real-time feedback and fine-grained control. Autoregressive generation, by contrast, naturally meets these needs: it integrates frame-level signals, supports diverse conditioning, and enables continuous interaction. By framing generation as a step-by-step prediction task, it promotes compact, causally grounded representations that are computationally efficient and well-suited to embodied agents. Revisiting autoregression is not a regression but a necessary progression—one that enables the development of practical, controllable, and intelligent world models. We contend that the future of video generation—and embodied AI more broadly—belongs to models that predict, compress, and revise the unfolding movie of the world, one actionable frame at a time.

## 2 World models

Before we dive into the discussion of video generation, it is worth pausing to examine why an agent needs a world model in the first place. A clear appreciation of this motivation will illuminate the position we have taken late.

At its core, a world model [5] is a compressed encoding of how the external world evolves in space and time. By internally rehearsing possible futures—compressing past experience while predicting what comes next—an agent can uncover the underlying physics and causal structure of its environment. This rehearsal acts as a dry-run for real exploration, a preview of an era where agents primarily learn from their own trajectories. Since an agent's sensorium provides an unlimited stream of training data, a scalable offline generative approach becomes essential: the model improves through imagination, not by risking failure in the real world.

Moreover, the human brain offers a compelling biological precedent for the necessity of world models [6, 7]. Our minds constantly engage in a similar "filling-in" mechanism, essentially constructing an internal model that mirrors the evolution of the external world to grasp its causal dynamics. This model functions as a continuous prediction engine, ceaselessly forecasting incoming sensory input and updating itself to minimize discrepancies between expected and actual outcomes. Concepts like the Bayesian brain and "controlled hallucination" further highlight this predictive and adaptive nature, where our brains aren't just passively receiving information but actively generating and refining a coherent internal narrative of reality through processes akin to dreaming and predictive coding.

# 3 Paradigm Shift of Video Generation

Over the past decade, video generation and prediction have undergone a clear paradigm shift—from early sequential modeling to today's full-sequence diffusion frameworks. Along the way, several initial assumptions proved flawed or misguided. Reflecting on these evolving perspectives offers both intellectual insight and essential context, laying the groundwork for the position advanced in the following sections.

## 3.1 The ConvRNN Era: Predicting Frame by Frame

Research into video generation began alongside the early successes of deep learning [8, 9, 10, 11]. Initially, researchers proceeded cautiously, working with simple datasets to test the feasibility of predicting future frames. In 2014, Srivastava et al. [12] first introduced a benchmark called Moving MNIST, derived from the MNIST dataset, and applied recurrent neural networks (RNNs), specifically LSTMs [13], for future frame forecasting. Building on this, Shi et al. [3] later proposed ConvLSTM, an architecture that combined convolutional operations with LSTM units. This allowed the convolutional layers to model spatial relationships while the recurrent layers focused on capturing temporal dynamics. Subsequent research extensively focused on improving this early framework. Some works proposed architectural enhancements to RNNs [4, 14], while others explored lossless compression techniques to compact video representations without sacrificing quality [15]. There were also efforts to incorporate variational inference to better capture the stochastic nature of future events, and methods that separately modeled foreground and background to enhance generation quality [16, 17].

Although Moving MNIST may seem trivial today, achieving strong performance on it was still challenging as late as 2020. At the time, models trained with Mean Squared Error (MSE) were known to produce blurry outputs, as MSE tends to average over multiple uncertain futures, merging distinct outcomes into one. Variational autoencoders (VAEs) [18], which compress visual data, were believed to worsen this blurriness. As a result, many video prediction frameworks avoided compression, opting for resolution-preserving designs. Meanwhile, the rise of GANs [19] in image generation heavily influenced video research. Many efforts integrated adversarial loss to reduce MSE-induced blur. While this led to sharper frames, it also introduced artifacts and disrupted temporal consistency. This highlighted a tradeoff: MSE models yielded blurrier but temporally stable videos, while GAN-based ones produced crisp yet often incoherent sequences.

## 3.2 The Diffusion Era: Denoising Step by Step

Before diffusion models rose to prominence, generative modeling had already begun shifting beyond traditional GANs, particularly in image generation. Two-stage pipelines emerged: a VAE or VQ-VAE [20] first learned to compress images into latent space, then an autoregressive transformer modeled that space with text conditioning. Early successes like VQGAN [21], DALL·E [22], and CogView [23] proved that models could reliably generate images aligned with arbitrary prompts—a major leap forward. Around the same time, diffusion models [24, 25] introduced a different generative approach: learning to reverse a noising process to turn noise into data via iterative denoising. Initially operating in pixel space with U-Net architectures [26], these models achieved impressive quality but at high computational cost. A turning point came with Stable Diffusion [27], which adopted a two-stage latent-based design, greatly improving efficiency and making high-quality image generation accessible on consumer hardware.

The natural next step was extending diffusion to video. Early efforts like AnimateDiff [28] and SVD [2] reused pretrained 2D VAEs and augmented U-Nets with temporal modules—typically convolutions or attention layers—to handle motion. These models applied holistic denoising over entire video latents, not frame-by-frame prediction. While promising, they often suffered from temporal artifacts like flickering, revealing the limitations of adapting image-centric architectures to dynamic video data. A major advance came with the Diffusion Transformer (DiT) [29] in Sora [1], which replaced U-Nets with transformers [30] operating on spatio-temporal patches, capturing long-range dependencies and improving motion consistency. Just as crucial was Sora's use of a 3D VAE, which compressed video into spacetime latents, enabling temporal compression and more coherent dynamics. Unlike 2D VAEs, training 3D VAEs requires distinct methods—e.g., using variable-resolution clips to enhance generalization. This methodological shift explains early reliance

on frame-wise 2D VAEs and underscores the growing consensus: temporal compression is vital for scalable, coherent video generation.

# 4 Driving Forces Behind Paradigm Shift

Having outlined the past decade's paradigm shift of video generation in the previous section, we now turn to an analysis of the key forces that drove this transformation. Understanding these underlying factors will set the stage for the following section, where we will argue for a return to autoregressive generation.

## 4.1 Why ConvRNNs paradigm fall short in video generation?

The decline of the ConvRNN framework stems from several limitations. First, its task formulation is inherently narrow: video prediction is typically framed as predicting the next $K$ frames from $N$ past ones. This restricts the model to short-term extrapolation based solely on prior motion. For example, in car-mounted videos, if the car is moving forward, the model will simply continue that motion, with no capacity for control. Even if given a command—like turning right—it won't know what should appear in the new view. Will there be a pedestrian or a tree? Resolving this ambiguity requires additional modalities to set expectations. Without them, the model faces high uncertainty and struggles to generate coherent long-term content. Worse still, ConvRNNs are poorly equipped to incorporate such multimodal signals, further limiting their usefulness.

Another key issue is their reliance on autoregressive generation. Since frames are produced one at a time, small artifacts introduced early in the sequence accumulate and intensify, degrading output quality—a problem known as compounding or drifting error.

## 4.2 The rise of text-to-video generation

The rise of diffusion models in video generation followed naturally from their breakthroughs in image synthesis — especially in text-to-image generation [2, 27], where they achieved high visual fidelity and strong compositionality. Building on this success, early video diffusion work focused on text-to-video (T2V) generation, directly inheriting both architecture and conditioning strategies from image diffusion models. But why has language — especially text — become the default conditioning signal for large-scale video pretraining?

Language is uniquely expressive and complete. It can precisely describe any concept, event, or entity, and its compositional nature allows for building complex ideas from simple components. Among all modalities, language alone enables both abstract generalization and fine-grained control [31, 32, 33]. In theory, one could even describe every pixel in every frame using natural language, suggesting a theoretical one-to-one mapping between text and video. This expressive completeness makes language a powerful interface for guiding generative models.

In video generation, pairing diffusion with text conditioning triggered a paradigm shift. Text naturally captures spatiotemporal events, aligning well with the holistic denoising process of diffusion. Unlike autoregressive models that predict frames sequentially, diffusion generates entire video sequences in latent space, promoting global consistency. Attention mechanisms support dense, bidirectional interactions across space and time, letting the prompt influence every region of every frame — ensuring semantic alignment with the input. As T2V dominates the field, diffusion has emerged as the leading video generation framework.

> **Full-sequence diffusion models is a natural fit to text-to-video generation.**
>
> The shift from ConvRNNs to diffusion models in video generation stems from the rise of text-to-video synthesis, where full-sequence diffusion models excel at generating globally consistent, prompt-aligned videos through holistic denoising and strong attention control.

# 5 The Autoregressive Renaissance

While the integration of T2V tasks with diffusion models has delivered impressive results, video generation remains imperfect. A major limitation is generation speed — reflecting the tradeoff between computational cost and video quality. Though less critical for creative applications, this becomes a bottleneck for real-time tasks like world modeling. Various acceleration methods (e.g., DDIM [34], DPM-Solver [35]) have been explored, but real-time feedback remains a challenge. Meanwhile, conditional control is only beginning to reveal its full potential. Language's unmatched versatility has, in some cases, overshadowed the value of other modalities. Consider camera motion [36, 37]: it can be conveyed via pose data or as text like "the camera pans left." Does this diminish the role of structured inputs? From a world modeling lens, clearly not — precise multimodal control is essential for grounded, interactive, temporally coherent simulation.

We argue that the next shift in video generation will be a return from holistic diffusion to autoregressive generation. This isn't just about speed — though autoregressive models are faster — it's about enabling fine-grained, frame-level control. More importantly, autoregressive generation promotes predictive compression: by learning to generate future frames from past ones, models internalize compact, structured world representations. This predictive structure allows world models to move beyond surface correlations and capture real causal dynamics. While "next-step" need not mean "next-frame," autoregression reintroduces interpretability, controllability, and a causally grounded approach to understanding the world.

## 5.1 Autoregressive Imagination for Interactive World Modeling

The visual world model takes the form of a video generator for one reason: to let agents visually imagine alternative futures — a key prerequisite for planning and interaction. But this imagination must be fast. Asking a model to render a 10-second video over several minutes isn't just inefficient — it breaks the principle of agency. Real-world environments are dynamic and unpredictable. No intelligent agent, robotic or biological, can afford to wait before acting. Interaction must be real-time and continuous, just like how humans instinctively shift gaze or reach in response to unfolding events.

This immediacy demands more than abstract, long-horizon speculation — it requires short-horizon, frame-by-frame anticipation grounded in the present. Language excels at conveying goals or imagined outcomes, but it's ill-suited for fine-grained, low-latency control. This reveals a key mismatch: text operates at a coarse temporal scale, while decisions unfold at much finer resolutions. Here, autoregressive generation emerges as the natural solution — offering the agility and responsiveness needed for intelligent agents.

Crucially, not all conditioning signals are equal — they vary in modality and temporal granularity. A text prompt might describe a five-second scene globally, while control signals like camera pose, joint angles, or keypresses update every frame. Static attributes like background layout or identity must remain consistent throughout. Current frameworks often stack these signals into fixed sequences to align with global prompts, but this sacrifices reactivity and flexibility. Worse, signals across timescales may conflict — e.g., a global prompt suggests panning right, while frame-wise pose indicates motion left. These mismatches confuse the model and destabilize internal representations. Ideally, local, real-time signals should override global ones — yet multi-timescale control remains poorly understood.

Autoregressive generation, by contrast, naturally supports temporal hierarchy. It generates videos one frame (or patch) at a time, allowing real-time feedback and causal integration of both high-level intent and low-level updates. Like a game engine, it's responsive, grounded, and adaptable. This frame-wise structure also supports continual learning — enabling the model to update its internal state on the fly, adapt to new environments, and improve without full retraining. If the goal is an interactive world model — one that supports planning, adaptation, and real-time decision-making — then the future lies not in static, offline diffusion, but in fast, flexible, autoregressive imagination.

## 5.2 Autoregression for Causality Learning

In its most straightforward form, autoregressive video modeling defines the generative process as a factorized distribution over frames or spatiotemporal tokens:

$$p(x_{1:T}) = \prod_{t=1}^{T} p(x_t \mid x_{<t}),$$

where $x_t$ denotes the video content (frame or token) at time $t$. This formulation forces the model to predict the future from the past, mirroring the causal flow of time. Unlike holistic denoising methods, which generate frames jointly and bidirectionally, autoregressive models must isolate which aspects of the past are predictive of future outcomes. This naturally encourages disentanglement of causal factors from statistical correlations, since only truly informative features support accurate prediction.

This sequential structure aligns with the dynamics of the real world, where observations emerge through state transitions governed by physical laws and actions [38, 39]. Autoregressive paradigms are well-suited to capture such structure, especially when paired with control-aware architectures. For example, modeling the system as a Markov process with hidden states $s_t$ and control inputs $a_t$, the model learns:

$$p(x_t \mid x_{<t}) \approx p(x_t \mid s_t), \quad \text{where} \quad s_t = f(s_{t-1}, a_{t-1}),$$

revealing that next-step prediction is not merely about extrapolation but about learning compact state representations that encode the rules of temporal evolution. Recently, a growing body of work has shown that video diffusion models often lack physical common sense — further reinforcing the necessity of learning causal structures.

## 5.3 Autoregressive Prediction as Compression

To illustrate how autoregressive future prediction functions as a form of compression, let us consider a large language model (LLM) as a concrete example. [40] (Note that the same line of reasoning can also be seamlessly extended to video generation.) Suppose Alice intends to transmit a massive dataset $\mathcal{D}$ of length $n$. At a given point in time, the first $t$ words, denoted as $x_1, x_2, \ldots, x_t$, have already been transmitted. Without loss of generality, we assume that the dataset's dictionary has a size of $m$. In the worst-case scenario, each word is one-hot encoded in binary, requiring $\log m$ bits per word. Thus, the total cost of transmitting the entire dataset $\mathcal{D}$ using this naive encoding method is given by:

$$C_0 = |f_0| + (n - t) \log m, \tag{1}$$

where $|f_0|$ represents the constant overhead required for transmission.

Certainly, more advanced compression techniques can be employed to reduce the transmission cost. However, recent research has revealed an intriguing insight: training an LLM specifically to predict the next token is itself a highly efficient method of data compression.

Since we already have the transmitted words $x_1, x_2, \ldots, x_t$, we can leverage the output of the LLM, which provides a probabilistic distribution over the next token:

$$P(x_{t+1}|x_{1:t}). \tag{2}$$

Using arithmetic coding, we can exploit this predictive distribution to reduce the number of bits required for transmission. The theoretical upper bound on the number of bits required for encoding a token is given by:

$$-\log P(x_{t+1} = x_{t+1}^*|x_{1:t}) + 1. \tag{3}$$

Thus, the total cost of transmitting the dataset under this predictive coding scheme becomes:

$$C_1 = |f_1| + (n - t) + \sum_{i=t}^{n} -\log P(x_{i+1} = x_{i+1}^*|x_{1:i}). \tag{4}$$

Here, $|f_1| + (n - t)$ represents a constant term related to the transmission overhead and the model's internal encoding scheme. The main term that determines the efficiency of compression is:

$$\sum_{i=t}^{n} -\log P(x_{i+1} = x_{i+1}^*|x_{1:i}), \tag{5}$$

This depends on the model's predictive capability. While no theoretical bound can be derived, it can be estimated empirically. Hoffmann's experiments showed that a well-trained large language model (LLM) achieved a 14-fold data compression rate. Moreover, larger models yield higher compression, suggesting they better capture statistical regularities. In contrast, the best text compression algorithm from the Hutter Prize achieves only an 8.7-fold ratio, highlighting the efficiency of predictive coding in neural networks.

In sum, autoregressive modeling thus serves as both a predictive and compressive mechanism, encouraging the world model to internalize causal structures rather than overfit to surface patterns.

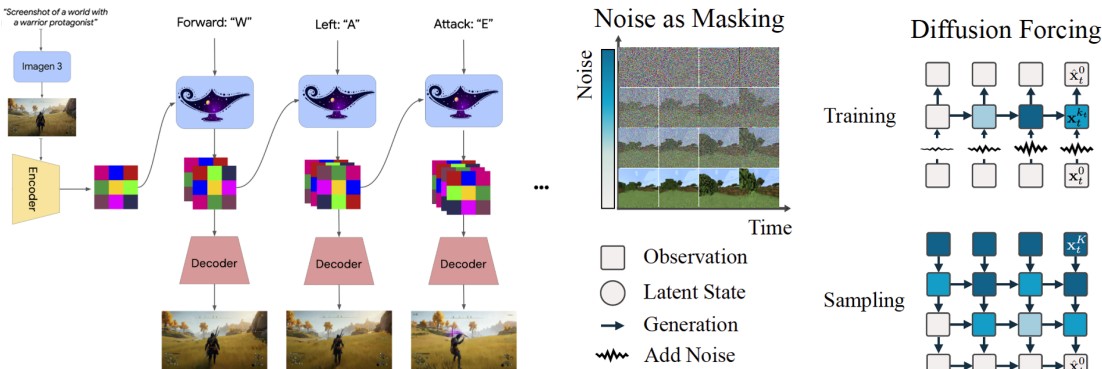

Figure 2: Autoregressive roll-out of Genie2          Figure 3: Diffusion Forcing denoising strategy

## 5.4 Escaping Language's Babysitting

Finally, we propose a speculative yet compelling question: in the relationship between language and visual world models, which truly comes first? While language can resolve much of the uncertainty in video generation, it may be better seen as a byproduct of our structured understanding of 4D reality. That is, cognition of space and time likely grounds language—not vice versa. [41]

Though language can be learned autoregressively without visual input, grounding it in visual experience may greatly improve sample efficiency. [42] This raises the possibility that the Text-to-Video paradigm may be somewhat inverted—we use language to guide video generation, rather than letting vision structure language. While language and vision likely co-evolved, overreliance on linguistic prompts may prevent video models from learning stable, intrinsic representations of the physical world.

Some argue that language serves as scaffolding—useful for structuring early learning. [43] But like scaffolding removed once a structure is complete, language may become less central. A mature world model may rely less on instruction and more on direct interaction—learning through perception, action, and embodied experience. Future paradigms should move beyond curated language data toward models grounded in raw experience.

> **Autoregressive generative models: the bedrock of the interactive world model.**
>
> Autoregressive video generation enables real-time, frame-level control and predictive compression, fostering causally grounded, interpretable world models.

## 6 Next-Gen Autoregressive Video Models

Having established the case for autoregressive generation as a foundation for world modeling, the next step is to explore how this shift can be practically realized. In practice, the reinforcement learning community, driven by a growing belief in the importance of world models, has already begun revisiting autoregressive video generation.

This section will therefore provide a concise overview and categorization of these nascent efforts, alongside proposing additional critical directions for future research. A central challenge in these early explorations stems from the quadratic computational cost of transformers — the basis for most state-of-the-art video generation models — as their cost increases with input length. As a result, much of the current research converges on a central question: how can we shorten latent sequences — or denoise only a small subset at each step — without compromising fidelity or control?

### 6.1 Next-frame Prediction Using diffusion

A simple yet effective way to implement autoregressive video generation is by adapting diffusion models to predict frames sequentially, generating videos one frame at a time using past latents and

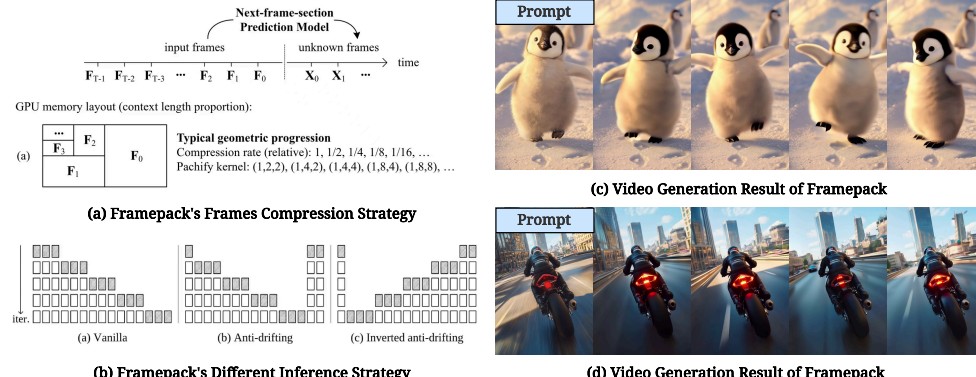

Figure 4: Illustration of Framepack: (a) One of the Framepack's past frames compression schemes. (b) Framepack's inference strategy, comprising causal autoregressive generation and two anti-drifting methods. (c) and (d) Example videos generated by Framepack.

actions. Genie 2 [44] follows this approach with a latent diffusion model combining an autoencoder and causal transformer to capture temporal dynamics, enabling framewise generation. Navigation World Model (NWM) [45] extends this to agent-centric navigation, using a Conditional Diffusion Transformer (CDiT) to simulate future egocentric views based on past inputs, modeling complex 3D transitions. GameFactory [46] applies this to games, injecting keyboard and mouse actions into pretrained diffusion models and supporting long-horizon generation via continuous conditioning and variable noise schedules. Together, these models show how diffusion can be repurposed for autoregressive, temporally coherent, interactive video synthesis.

## 6.2 Selective Denoising

Selective denoising—where video segments are denoised incrementally rather than in a single global step—has emerged as a scalable, flexible alternative to traditional diffusion-based generation. Diffusion Forcing (DF) [47] exemplifies this by treating generation as partial unmasking: each token is noised independently, and the model learns to denoise arbitrary subsets using a shared next-token prediction setup. This enables flexible sequence lengths, adaptive schedules, and compositional generalization. Oasis [48] applies DF to complex settings like Minecraft, generating coherent frames conditioned on evolving states. Similarly, MAGI-1 [49] uses chunk-wise autoregression with diffusion transformers for temporally consistent, stepwise video generation. These methods showcase adaptive denoising's benefits—greater temporal flexibility, lower compute, and enhanced interactivity—making it well-suited for real-time applications.

## 6.3 Adaptive Resolution

Another valid strategy to improve autoregressive video generation is to enable adaptive resolution. FramePack [50] achieves this by compressing past frames into a fixed-size memory, preserving only the most salient information while discarding redundant details. This allows autoregressive models to maintain constant compute regardless of sequence length, effectively acting as a temporal resolution filter. Framepack's compression and inference strategy and generated samples can be seen in Fig. 4. If extended to video, models like VAR [51] could adopt similar strategies—using resolution-aware tokens or dynamic memory to scale autoregressive generation more efficiently. Inspired by human saccadic vision, where only the gaze center is rendered in high resolution, future autoregressive models might employ foveated rendering, focusing compute on regions of interest and blurring others—achieving adaptive resolution in both time and space.

## 6.4 Postdictive Coding

Most video generation research emphasizes predictive coding—inferring future states from prior observations [52]—while largely neglecting postdictive coding [53], the process by which new sensory

input reshapes interpretations of earlier events. This retroactive revision is vital for autoregressive, online world models, where predictions must be continuously updated as new data arrives. When mismatches occur between earlier forecasts and later inputs, models need to revise latent memory to preserve consistency. Cognitive science supports this: effects like the flash-lag and cutaneous rabbit illusions show how the brain integrates sensory information over time to construct coherent, post-hoc perceptions. Yet despite its biological relevance, postdictive coding is mostly absent from current video generation frameworks, which typically use rigid feedforward designs—where the past influences the future, but not vice versa. This limits real-time adaptability and self-correction. Incorporating postdictive mechanisms could refine past representations with future evidence, enhancing causal reasoning and temporal coherence. A hybrid system combining predictive foresight with postdictive revision promises more adaptive, cognitively grounded video models—capable of learning not only by anticipating what comes next, but by reinterpreting what came before.

# 7 Alternative Views

**View #1: If autoregressive generation was once abandoned, why return to it now—and what makes this time different?**

The limitations of early autoregressive video generation were not due to flaws in the paradigm itself, but external constraints. The dominant task—predicting future frames from short histories—ignored multimodal control, not from oversight, but due to missing interfaces, limited paired video data, and the lack of strong generative backbones. Today, these barriers are dissolving. With growing interest in fine-grained control and rich conditioning, autoregressive modeling is being revived—not as regression, but as a path to real-time, interactive world simulation.

**View #2: Autoregressive generation still suffers from compounding errors—how do we address that this time?**

Compounding error has long challenged autoregressive video generation, stemming from the accumulation of small imperfections over time. Earlier models often produced blurry frames or artifacts that, when recursively reused, degraded quality. Yet this issue isn't inherent to autoregression—it depends on the fidelity and temporal consistency of predictions. When outputs are clean and coherent, error accumulation becomes far less problematic.

Text conditioning further narrows the space of possible futures, injecting semantic structure and reducing uncertainty. Importantly, revisiting autoregression today doesn't mean abandoning text guidance—it signals a move toward uniting fine-grained temporal modeling with high-level intent. Thanks to improved architectures, richer training data, and multimodal inputs, the factors that once amplified compounding errors have been largely addressed.

**View #3: If my sole purpose for video generation is to create creative content, does the return to autoregressive models still not concern me?**

Yes—autoregressive generation offers practical advantages beyond world modeling. Most notably, it enables faster inference, giving creators near-instant feedback during creation. This speed accelerates iteration and makes interactive, game-like content possible, pushing beyond static video playback into dynamic, engaging media.

# 8 Conclusion

We have taken the position that video generation, particularly in the context of world modeling, stands to benefit significantly from a return to autoregressive paradigms. While diffusion-based models have excelled at holistic scene synthesis and compositional generalization, they fall short in scenarios that demand real-time responsiveness and precise control. Autoregressive generation, in contrast, is inherently better suited for integrating frame-wise signals, accommodating multimodal conditions, and adapting to continuous feedback—all of which are essential for interactive and causally coherent simulation. By framing generation as a sequential prediction task, autoregressive models promote compact, causally grounded representations that are not only faster to compute but also more aligned with the needs of embodied agents. As we look toward building practical, interpretable, and controllable video models for future world-simulating systems, we believe the autoregressive approach offers the most promising foundation.

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
