# OpenReview forum: "Spiral Evolution of Visual World Model: Reclaiming Autoregression from the Diffusion Era"
_NeurIPS.cc/2025/Position_Paper_Track — Submitted to NeurIPS 2025 Position Paper Track_

### Official Review · Reviewer_Fyfd · 2025-08-07

**Significance:** 3
**Presentation:** 4
**Rating:** 8
**Confidence:** 4

**Summary:**

This paper argues for a renewed focus on autoregressive architectures as the future of video generation and world modeling. The paper argues that while current diffusion models excel at visual fidelity and prompt adherence, their reliance on global optimization and heavy compute resources makes them unsuitable for real-time applications that demand fast, coherent, and causal generation. In contrast, the paper proposes that the research community revisit autoregressive architectures. These models offer more efficient underlying representations that support better temporal coherence and allow for fine-grained control during the generation process. Finally, the authors show that this approach could serve as a more viable path forward for building interactive, visual artificial intelligence systems in the future.

**Strengths:**

I think one of the core strengths of this paper is that it is structured very well – the paper introduces the concept of world models and then guides the reader through the history of autoregressive model architectures followed by the recent advances made in video diffusion models. This cohesive flow, coupled with strong literature references, allows the paper to develop its position supporting the revival of the autoregressive architecture by demonstrating to the readers why current approaches fall short and how recent developments can make these autoregressive architectures both viable and more powerful. Furthermore, I strongly believe that this perspective can be very useful to the computer vision community at NeurIPS, particularly in scaling future world/video models more efficiently.

**Weaknesses:**

One very minor weakness I’d like to point out is that, while the paper mentions postdictive coding—which involves refinement through revision—it may be somewhat confusing to readers. This is because it could be interpreted as endorsing bidirectional (temporal) interactions between representations, which appears to contradict traditional autoregressive architectures that strictly avoid relying on future representations. Some clarification on this point would be great.

**Questions:**

I think another interesting (though slightly tangential) question to consider is how switching to an autoregressive model architecture would affect model size compared to traditional diffusion models?

**Alternative Position:**

Yes, and alternative positions are well-considered and addressed by the argument

**Author Identification:**

No.

**Context:**

4

**Discussion:**

4

**Ethics:**

["NO or VERY MINOR ethics concerns only"]

**Position:**

Yes, the paper argues for or against a position related to machine learning.

**Support:**

4

**Thoroughness:**

4

---

### Official Review · Reviewer_BLNB · 2025-08-07

**Significance:** 3
**Presentation:** 3
**Rating:** 4
**Confidence:** 4

**Summary:**

The paper advocates for a revival of autoregressive models in video generation, particularly for interactive world modeling, contending that while diffusion models achieve impressive visual fidelity and global coherence, they are ill-suited for real-time, control-intensive applications. Autoregressive approaches, with their sequential generation, provide several key advantages: they ensure causal coherence by aligning with natural temporal dynamics, enable fine-grained control and deliver real-time responsiveness—a critical requirement for embodied agents operating in dynamic environments. By leveraging these strengths, the authors argue that autoregressive models are better positioned to meet the demands of interactive and responsive world simulation compared to their diffusion-based counterparts.

**Strengths:**

- The paper presents a valid argument for revisiting autoregressive models in video generation, particularly for world modeling. This contrasts with the current dominance of diffusion models, offering a fresh perspective that aligns with the needs of real-time, causally coherent applications.

- The authors provide a thorough historical overview of video generation paradigms, from ConvRNNs to diffusion models, and critically evaluate their limitations. This contextual grounding strengthens their argument for an autoregressive renaissance.

**Weaknesses:**

While the advantages of autoregressive approaches are well-articulated, the discussion overlooks key innovations that have significantly improved diffusion models in the very areas where they were previously weak. For example:

- real time models: techniques like latent consistency models (LCMs) [1] and distillation methods now enable near real-time generation without sacrificing quality.

The arguments for autoregressive models are valid, but the paper would benefit from an up-to-date comparisons with modern diffusion techniques.


[1] https://arxiv.org/abs/2310.04378#

**Questions:**

Please see weakness.

**Alternative Position:**

Yes, and alternative positions are well-considered and named but not addressed

**Author Identification:**

No.

**Context:**

2

**Discussion:**

3

**Ethics:**

["NO or VERY MINOR ethics concerns only"]

**Position:**

No, the paper presents new research without clearly advocating a position.

**Support:**

2

**Thoroughness:**

4

---

### Official Review · Reviewer_NK7N · 2025-08-13

**Significance:** 2
**Presentation:** 2
**Rating:** 3
**Confidence:** 4

**Summary:**

This paper advocates revisiting the autoregressive paradigm as a foundation for building interactive world models. This paper summarizes some technical directions in the field of video generation from past ConvRNNs to nowadays diffusion models. By demonstrating some disadvantages of diffusion models for interactive video generation, this paper advocates that the research community should focus on autoregressive generation. Several advantages of autoregressive generation are analyzed to support the paper's position.

**Strengths:**

- This paper considers a widely-discussed problem, whether to use autoregressive or full-sequence generation in the context of interactive video generation. This topic is important and valuable to the research community.

**Weaknesses:**

- The figures used in this paper are borrowed from other papers, such as Figures 2~4, which raises concerns regarding the originality of the visual content.
- The evidence provided to support the paper's position is not sufficient or strong. This paper claims that autoregressive generation is better suited for interactive generation due to the time efficiency, while diffusion models require more time. However, for generating a long video, a diffusion model with a faster sampling technique (e.g., fastvideo) can surpass autoregressive generation, especially when a frame requires lots of tokens to reconstruct. The advantages of autoregressive generation demonstrated in section 5 should be further highlighted, and it should be explained clearly why other approaches fail to do so.
- The paper's position and writing are not clear. While criticizing diffusion models in the introduction section, the authors propose to leverage diffusion for next-frame prediction in section 6.1.
- No novel directions or insights are provided in this paper. All the discussed points have been mentioned in previous research, at least for me.

**Questions:**

The compounding error in autoregressive generation still poses a challenge for high-quality generation. Do you have any idea for addressing it?

**Alternative Position:**

Yes, and alternative positions are well-considered and named but not addressed

**Author Identification:**

No.

**Context:**

2

**Discussion:**

3

**Ethics:**

["NO or VERY MINOR ethics concerns only"]

**Position:**

Yes, the paper argues for or against a position related to machine learning.

**Support:**

1

**Thoroughness:**

3

---

### Note · Authors · 2025-09-05

**1-11 Submit Again:**

Probably no

**1-1 Submission Process:**

3

**1-2 Next Year:**

We are deeply grateful to the PC and AC for their thoughtful efforts in organizing this year’s NeurIPS position paper track. Your dedication has created a valuable platform for exchanging forward-looking ideas and shaping the field’s future directions.

Here are our expectations for next year’s Position Paper Tracker:

1. Introduce a rebuttal process similar to that of the main conference.
We believe that the concerns raised in this review could be addressed fairly straightforwardly if a rebuttal process were available.


2. Provide reviewers with a more detailed guideline for evaluation standards.

**1-4 Interest:**

["Panel discussions with other position paper authors", "Structured debates on controversial topics"]

**1-5 Thoughtful:**

3

**1-6 Supportive:**

5

**1-7 Technical Aspects Versus Position:**

5

**1-8 Gate Keeping:**

3

**1-9 Camera Ready Changes:**

We plan to make the following revisions in the camera-ready version:

1.  Add More Latest Works: With the emergence of several works Genie3 [1], Mirage [2], Matrix-Game [3] in August confirming the trend of autoregressive video generation anticipated in this position paper, we find it appropriate to incorporate a discussion of these new developments in Chapter 6.

2. Additional Section on Sampling Acceleration Techniques: The reviewer noted that our position paper does not discuss recent advances in sampling acceleration techniques for diffusion models. While this is briefly mentioned in line 165, we plan to add a new section in Chapter 6 of the camera-ready version to introduce these methods. We believe, however, that this comment arises from a misunderstanding. Sampling acceleration techniques are not inherently in conflict with the autoregressive video generation paradigm. In Section 5.1, we may need to clarify more explicitly what “fast” means in this context: it refers both to the generation speed and to the short-horizon nature of predicted futures. Generating long-horizon futures in a single shot is often less relevant for building interactive world models. Although acceleration techniques can improve generation speed, they remain relatively independent of the autoregressive framework. Therefore, including this discussion does not alter our central position. Nonetheless, adding such a section would still be valuable for readers with an interest in this area.

3. Add the Spatial Memory Subsection: In chapter 6, we had initially included a subsection on spatial memory. This was omitted from the submission due to page limits. If the camera-ready version permits additional pages, we plan to reinstate this subsection, as it offers valuable context for guiding future model designs.


[1]. https://deepmind.google/discover/blog/genie-3-a-new-frontier-for-world-models
[2]. https://blog.dynamicslab.ai/
[3]. https://huggingface.co/Skywork/Matrix-Game-2.0

**3-1 Review Response1:**

NK7N

**3-2 Reaction To Review1:**

We thank the reviewer and believe our clarifications will further strengthen the paper’s contribution. We believe the points raised can be readily addressed as follows:

1.  In position papers, reusing figures is a widely accepted practice to clarify ideas and assist readers, rather than to claim novelty. For example, two ICML 2024 position papers on generative models [1], [2] incorporated figures from published works.

2. Whether or not sampling acceleration tricks are used does not affect our argument for autoregressive generation. They can be combined. This was already noted in line 165, but the reviewer seems to have misunderstood Section 5.1. Here, “fast” refers both to generation speed and to the short-horizon nature of predicted futures, as long-horizon holistic generation is less relevant for interactive world models. Since acceleration methods are fully compatible with the autoregressive framework, this discussion does not alter our position. Still, adding such a section could benefit interested readers and can be easily addressed.

3. We believe this is another misunderstanding. At no point in the paper do we oppose diffusion models. What we contrast with autoregressive generation is instead full-sequence generation. While prior paradigms emphasized producing entire videos holisticly, we argue that real-time autoregressive prediction is more appropriate for building interactive world models.

4. It also seems the paper may not have been read with sufficient care. Points such as revisiting ConvRNN-era autoregression, the success of the diffusion era stemming from the complementarity of T2V and full-sequence generation, postdictive coding (asked by Reviewer Fyfd), saccades, moving beyond language control, and the arguments for the necessity of autoregressive generation are all novel contributions of our position paper, newly introduced in this domain.

[1]. https://openreview.net/pdf?id=SoNexFx8qz
[2]. https://openreview.net/pdf?id=EZH4CsKV6O

**3-3 Review Response2:**

BLNB

**3-4 Reaction To Review2:**

We thank the reviewer for recognizing the strengths of our paper and for the constructive suggestions, as well as for acknowledging that the arguments for autoregressive models are valid. We believe our clarifications will further strengthen the paper’s contribution.

1. Sampling acceleration:  We believe this comment reflects a minor misunderstanding of our position. The use of sampling acceleration techniques does not affect our argument for autoregressive generation; the two are fully compatible and can be applied together. And it was briefly mentioned in line 165.

In Section 5.1, “fast” refers not only to generation speed but also to the short-horizon nature of predicted futures, which is critical for building interactive world models. While acceleration methods can certainly enhance generation speed, they remain orthogonal to the autoregressive paradigm. Therefore, including a short section on this topic would not alter our central position. Still, adding such a section could benefit interested readers and we are happy to incorporate it in the camera-ready version.

**3-5 Review Response3:**

Fyfd

**3-6 Reaction To Review3:**

We sincerely thank the reviewer for their encouraging feedback and recognition of our paper’s clear structure, strong literature grounding, and the value of our perspective on autoregressive paradigm. We are also grateful for the acknowledgment that our arguments are valid and relevant for scaling future world and video models. We believe our clarifications will further strengthen the paper’s contribution.

1. Postdictive Coding: Your understanding is very accurate. The possible implementation of postdictive coding would result in a mixed regime between bidirectional interactions and autoregressive generation. This area is still underexplored, as current full-sequence and autoregressive models are not designed to incorporate real-time ground truth updates during generation.

Once real inputs are accepted at inference time, the intervals between successive ground truth observations become periods of pure model imagination. Even if generation within these intervals is autoregressive, it remains simulated and should be overridden by real inputs when conflicts arise.

While the implementation details are still open, we believe this kind of real-time interaction—balancing simulation with incoming real data—is essential for effective world models and learning from experience.

2. Model size: This is also a very interesting question, and there are many perspectives from which one could approach it. I will briefly discuss just a couple of points here:

a. Whether to use a T2V pretrained model: If such a model is used, the model size will typically be comparable to that of the pretrained version.

b. Whether to incorporate explicit spatial memory: If used, the learning burden on the model may be reduced, potentially leading to a smaller parameter count. More specifically, the structure of the memory module used can have a significant impact on both performance and model complexity.

---

### Meta-Review · Area_Chair_BBXF · 2025-08-31

**Rating:** 4
**Confidence:** 4

**Strengths:**

This paper puts the spotlight on video generation. It emphasizes the limitations of diffusions models and the potential advantages of autoregressive models. This is particularly important in for the time-sensitive world models.
Video models and world models are of high interest to NeurIPS attendees and so the general topics may inspire discussion.
Some reviewers appreciated the historical overview and found that it gave motivation to the argument.

**Weaknesses:**

This paper was criticised for not providing enough new insights, but is mainly recalling the hitory of past models. This, by itself, might not be interesting enough for NeurIPS attendees.
The paper can put more effort in discussing improved / faster diffusion approaches.
Diffusion has recently been shown to be very fast for generating long sequenes. This suggests there is hope yet for diffusion models for video. There is not much discussion of that here.
It is not 100% clear what the paper is advocating for, as autoregressive and diffusion generative models are not mutually exclusive.

**Questions:**

What is your position on combining Ar and diffusion models for video?

**Thoroughness:**

3

---

### Decision · Program_Chairs · 2025-09-26

Reject